# Impact Study of Temperature on the Time Series Electricity Demand of Urban Nepal for Short-Term Load Forecasting

Yaju Rajbhandari [1,*], Anup Marahatta [1], Bishal Ghimire [1], Ashish Shrestha [2], Anand Gachhadar [1], Anup Thapa [1], Kamal Chapagain [1] and Petr Korba [3]

[1] Department of Electrical and Electronics Engineering, Kathmandu University, Dhulikhel 45200, Nepal; anup.marahatta@ku.edu.np (A.M.); bishalghimire1997@gmail.com (B.G.); anand.gachhadar@ku.edu.np (A.G.); anup.thapa@ku.edu.np (A.T.); kamal.chapagain@ku.edu.np (K.C.)

[2] Department of Electrical Engineering, Information Technology and Cybernetics, University of South-Eastern Norway, N-3918 Porsgrunn, Norway; sthashish2010@gmail.com

[3] School of Engineering, Zurich University of Applied Science, DH-8401 Winterthur, Switzerland; korb@zhaw.ch

* Correspondence: yaju.rajbhandari@ku.edu.np or rajbhandari.yaju095@gmail.com

**Abstract:** Short-term electricity demand forecasting is one of the best ways to understand the changing characteristics of demand that helps to make important decisions regarding load flow analysis, preventing imbalance in generation planning, demand management, and load scheduling, all of which are actions for the reliability and quality of that power system. The variation in electricity demand depends upon various parameters, such as the effect of the temperature, social activities, holidays, the working environment, and so on. The selection of improper forecasting methods and data can lead to huge variations and mislead the power system operators. This paper presents a study of electricity demand and its relation to the previous day's lags and temperature by examining the case of a consumer distribution center in urban Nepal. The effect of the temperature on load, load variation on weekends and weekdays, and the effect of load lags on the load demand are thoroughly discussed. Based on the analysis conducted on the data, short-term load forecasting is conducted for weekdays and weekends by using the previous day's demand and temperature data for the whole year. Using the conventional time series model as a benchmark, an ANN model is developed to track the effect of the temperature and similar day patterns. The results show that the time series models with feedforward neural networks (FF-ANNs), in terms of the mean absolute percentage error (MAPE), performed better by 0.34% on a weekday and by 8.04% on a weekend.

**Keywords:** energy consumption; short-term electricity demand forecasting; feedforward neural network; temperature impact on electricity demand

## 1. Introduction

The total number of households having electricity access in Nepal has reached 86% in the year 2020, and the consumption of electricity has increased from 6303 GWh to 6422 GWh over the past year. With expanding supplies and growing demand in the country, the only electric utility company in Nepal, the Nepal Electricity Authority (NEA), has practiced a load forecasting technique to implement proper planning and predict the required generation until fiscal 2039/40 to be 82,620.73 GWh [1]. Currently, the Nepal Electricity Authority is focused on increasing the consumption of domestic demand through the promotion of electric vehicles (EVs), increasing lift irrigation facilities, and replacing fuel-based cooking with electricity. With the increment in demand in urban and industrial areas, the expansion and improvement of transmission and distribution infrastructures are continuously being carried out with a high priority in the country [1]. With the exponentially growing demand, the energy balance becomes complex. As the Nepalese electricity market is moving from being supply-based to being demand-based, fulfilling

demand becomes very important. Hence, understanding the growth and variation in load becomes a significant aspect of the sustainability of the country's energy sector.

Forecasting of electric load is one of the best ways to understand the present condition and future probability of the load. The practice of load forecasting in Nepal is focused on long-term purposes only [1,2]. As for knowledge and publications, only a single attempt has been made at short-term load forecasting of load for Nepal. The paper used an artificial neural network (ANN) to forecast the load of Kathmandu Valley [3]. The model developed utilized multiple variables such as the temperature, hour of the day, day of the week or weekend, and weekday indicators to complete the forecast. The research work mainly focused on utilizing an ANN to forecast load, yet it failed to explain the relationship between the variables and the load data. Even though the model presents a satisfying forecasting output, understating how the variable selected affects electricity demand becomes important, which is the major focus of this paper. Kathmandu is the capital of Nepal, and the demand for electricity in this area tends to be the highest in the country, ranging from 101.12 to 229 MW throughout the day [3]. This variation throughout the day is due to consumers' daily activities and their reactions to the effects of environmental and social factors. Understanding these effects allows the Nepal Electricity Authority to better plan and even implement demand response to manage it in the future [4]. The literature on the study of load around the world shows that understanding these relations with influential factors plays a major role in clarifying the demand fluctuation and helps develop better load forecasting models for electric load [5].

Short-term load forecasting (STLF) has become an important subject to energy suppliers and system operators for efficiently operating electric power systems. A robust STLF model allows for tracking the hourly load variation of load for the day or even week ahead with high accuracy. In developed countries, electricity distribution is fixed to a day ahead, which in real time is managed using demand-side management strategies, resulting in the reduction of high operation costs. However, a lack of accuracy in load forecasting can have a huge impact on system reliability [6]. The complex behavior and fluctuating nature of electricity demand is a major challenge when developing a robust model. Fluctuation in the consumption of electricity is due to the random behaviors of the consumers, but within all the randomness, a few predictable patterns can be separated, such as seasonal, weekly and daily variation [7]. As human behaviors and requirements change with various factors such as the time of day or the day of the week, holidays tend to follow similar demand patterns, which make them easier to track [8]. These deterministic variables show a major impact on the electricity demand, especially during special days. As such events result in people traveling, most people travel away from major cities largely during long holidays or religious holidays. Local festivals and major events such as the new year have a huge impact on the load consumed [7]. This effect can be observed in various locations, which affects the consumption pattern [9,10]. Understanding these special days can add to improving the forecasting accuracy of the load. For short-term load forecasting, the hourly influence has a significant effect on the load such that variations in consumption during the morning hours, day time, and night hours show a daily demand pattern of electricity. Likewise, stochastic variables such as environmental factors and human behaviors toward the change in the environment also provide information on changing electricity demand. Among all factors, the outdoor temperature shows an important influence that explains the variation in demand [7,11,12]. The influence can be linked to the use of electric cooling and heating appliances used within the households. The literature shows that a change in the outdoor temperature results in an increase or decrease in the use of appliances, resulting in a change in electricity demand. As multiple examples of research show that heating and cooling appliances are responsible for more than half the energy consumption in the residential load [7], implementing the impact of temperature as an input to the forecasting model can improve the accuracy of the load forecasting model.

Being the most populated city in Nepal, Baneshwor is the center of the capital city of Nepal, comprising major human activity and an official workstation. As a developing

country, centralization in the city is a major issue, which results in increasing demand. In such conditions, keeping the supply intact within the valley becomes important to maintain day-to-day activities. To avoid any power surges or loss, a proper study of the load should be performed. Forecasting the load for each hour of the day allows the Nepal Electricity Authority to avoid any major changes to the system which may cause a blackout, load shedding, or voltage fluctuation in the system. Therefore, the main objective of this paper is to present an accurate short-term load forecasting model and perform an evaluation of the impact of electric load influencing factors that affect the consumption of electricity. Based on the literature, this paper is the first to present a detailed study of the variables affecting electricity load consumption within Nepal. With the study of electricity demand in the Baneshwor substation and its relation between the previous day's load of the same hour and temperature and an environmental variable affecting the load, this paper presents an ANN model which forecasts load for the next 24 h. The major findings based on the electricity data for STLF are explained as follows:

- An impact study on the temperature is performed, which leads to increases in electricity demand during hot and cold temperatures for a tropical region like Nepal.
- The quantitative analysis among variables such as the impact of time series variables such as holidays, working days, the time of day, day of the week, and day of the month is discussed in detail.
- The development of two different forecasting models for weekends and weekdays, which leads to achieving better prediction capability among the existing literature of the Nepalese dataset.
- A comparison of developed models with conventional time series models is conducted for weekdays and weekends, which shows that the FF-ANN holds the upper hand when forecasting load for both days, with a major improvement in weekend forecasting.

Furthermore, this paper is organized as follows. Section 2 discusses the literature review and past work of load forecasting. Section 3 discusses the data used for thorough analysis of load relation with the lag load and temperature. Section 4 presents the methods implemented for the forecasting background and parameter selection. The results from forecasting using different methods are discussed in Section 5, and Section 6 is the summarization of this paper's findings.

## 2. Literature Review

Electricity load forecasting is a prediction evaluation technique that is conducted mostly by the utility power company to find the tentative demand for energy for the future. It allows the utility company to understand load growth and fluctuation, which helps the suppliers manage their generation according to the demand of the end users. Forecasting has become an important process in the planning and operation phases of the system, allowing energy companies to have a better understanding of the demand characteristics. Load forecasting is conducted at different levels, such as the long term, medium term, and short term [5]. Depending on the requirements, forecasting is conducted for the long term, covering yearly load demand prediction, the medium term forecasts the load between one week and months, and short-term forecasting ranges between one hour and one week. However, demand forecasting is sensitive to various aspects, ranging from previous load consumption to socioeconomic factors, as well as changes in weather conditions. Variable selection for performing the forecast is a significant part of the process. In terms of the long-term forecasting of load for a country, the forecast considers the domestic production, population change, import–export trade, overall weather, and economic change observed in that country as major variables [13]. A study performed for long-term energy consumption forecasting included the gross domestic production (GDP), population, import–export trade, and primary energy consumed per year and forecasted the consumption pattern [13]. Likewise, medium-term load forecasting is applied to identify maintenance scheduling, coordinate supply stations, manage energy units, and develop cost-effective energy purchase

strategies [14], whereas short-term forecasting focuses on identifying the hourly variation of demand such that the study observes the day-to-day energy trends, seasonal effects, temperature variations throughout the day, and day-to-day consumer activities [7,8,10].

Model selection is another important part of the process for the better forecasting of load. To maintain accuracy, various approaches have been in practice in forecasting studies, such as the multiple linear regression model [8], autoregressive integrated moving average model (ARIMA) [15], exponential smoothing [16], deep learning including artificial neural networks [17], recurrent neural networks, particle swarm optimization (PSO), and hybrid techniques [5]. Among the various models being used, deep learning has gained much attention recently [7,15,17]. This approach of the machine learning model breaks through the constraints of the original mathematical calculations in terms of accuracy. It realizes and identifies the pattern within the data through neural networking and is better at tracking the nonlinearly within the data [13]. However, training the model with proper parameters and constraints is the challenging part of neural network modeling. For better tracking, the patterns within the data hybrid model combined with the neural network hold the upper hand, as the comprehensive learning particle swarm optimization (CLPSO)-based memetic algorithm (CLPSO-MA) shows evolved feature selection and parameter optimization simultaneously [18]. Similarly, the authors of [19] implemented particle swarm optimization as a training algorithm to adjust the weights of artificial neural networks, where the particle swarm algorithm has shown its ability to minimize the error function with maximum efficiency. These models of forecasting demand great computation and programming skills, such that data are passed through a black box for processing. However, the conventional approach [8] of prediction that uses a mathematical approach presents a satisfying level of accuracy. The study classified the hourly load data through the statistical approach for identification of an optimal mode of prediction. Simple methods such as a simple average of the previous four weeks performed quite well based on the relation with previous lags. A load model developed using autoregression methods and a Holt-Winters-Taylor exponential smoothing method tended to outperform other models [20]. With proper analysis, simple and conventional methods can save time as well as computation power while forecasting.

In short-term load forecasting, accuracy becomes a crucial factor, since it is used to optimize the day-to-day operational efficiency of electrical power delivery [21]. The forecasting is done on an hourly and day-to-day base prediction of load precision data, which drives the decision on the load and supply for the next day or hour. This prediction method plays a key role for power companies, as they need to develop load plans and short-term strategies in particular, which has attracted considerable attention particularly for building power management to avoid voltage fluctuations and blackouts [22]. The forecasting models vary within short-term load forecasting (STLF) [23] and classify the methods in four categories: similar pattern, variable selection, hierarchical, and weather station selection. It finds the similarities within the data such as the time, day, and weather conditions of the load profile to predict the future load of the specified time, whereas the variable selector selects the most suitable variable within the dataset while they can adequately capture the relationship between the available input data and the output (i.e., electrical demand). The models developed within these concepts identified the effects of the variables on the output, such as monthly effects, daily to weekly effects, weather effects as well as social and local effects. Similarly, the authors discussed the hierarchical methods, such as bottom-top and top-bottom, which showed load studying from the appliance level to the utility level [23]. Forecasting can be done at a different level to decide specific levels that may improve the accuracy of the results. Based on the available data, proper model selection must be done, as a huge dataset does not mean better forecasting, such that the selection of the wrong model for the selected dataset or selection of the wrong dataset can decrease the model's accuracy. Within a variety of models, current research has been conducted to identify the best forecasting model. One study [17] used a combined genetic algorithm (GA) with machine learning to choose the best features to be

used for forecasting and finally used long short-term memory (LSTM) to obtain the best predicted output. Likewise, the authors of [7] combined multiple linear regression (MLR) with machine learning to perform short-term load forecasting, considering the MLR model to establish a relation with the input variables.

### 2.1. Temperature Effect

The short-term electrical load variation can be related to factors like the environment, population, or economy. Studies indicated a direct relation of load with temperature [7,11,12]. A study performed in France showed that the temperature has a negative correlation of 0.94 with the load [22], whereas a study performed in Bangkok's metropolitan area found that the electricity demand for residential load tended to increase by 6.79% for a 1 °C rise in temperature [24]. These two studies indicate a very different type of relationship for different locations based on a topological effect. Establishing a relation can be a challenging task. In the case of France, the country lies in a region with mild temperatures, such that the temperature varies from −5 to 25 °C, whereas Bangkok lies in the equatorial region, and the temperature varies from 25 °C in the winter to 31 °C in the summer. Both have different temperature ranges, where one has cold temperatures and the other country has warm temperatures throughout the year, and electricity demand in both regions increases with an increase or decrease in temperature. The effect of the temperature may change from country to country, based on their topology and geographical location such that the effect can be positive, negative, or both in some cases. In Bangladesh, the temperature effect is fitted using a linear relation with the load [25]. Research shows that the linear slope of the electric load and temperature is higher during the summer rather than the winter season, as the rate of load increment is 53.14 MW/°C during the summer and 2.14 MW/°C during the winter. One study conducted on the electricity demand of China explored the relation by dividing the sample into two groups, south and north, which explained the effect of the temperature in different topologies. The results showed that the load increased by 0.002% for every one-degree decrease in temperature for the southern region, whereas the load in the north region tended to use 0.021% more electricity than the southern region with every one-degree increase in temperature [26]. For the load dataset of Thailand, the variation was discussed in terms of hours. The load varied from 50 MW/°C during the morning and evening hours, and the impact of the temperature was much higher during nighttime hours, where the results showed an increment of 100–200 MW/°C [7]. A similar case study conducted on Hokkaido showed that the rise in temperature from 25 to 26 °C increased the demand by 7% during daytime hours, whereas there was only a 2% increase during nighttime hours [27].

However, depending on the type of consumers, the relation with the temperature can be different. A study performed on European load indicated that the demands for commercial and industrial loads were not affected much by the variation in temperature [11] such that the commercial and industrial loads had somewhat constant demand when considering the temperature. In the case of Hong Kong, the temperature tended to have a higher effect on the residential load in comparison with the commercial and industrial loads [12]. The analysis on the electrical load showed a 9.2% increment with the temperature for the residential sector, whereas the load only increased by 2.2% and 3% in the commercial and industrial sectors, respectively. These findings can be related to a load study done for Pune, India. The data showed that the industrial activities were not affected much by temperature variations, whereas residential activities showed an increment in the load of 1.5–2%/°C change in temperature [28].

### 2.2. Time Series Dependency

The electrical load shows good correlation with previous load data. Observing the change in load and other variables can be linked to the electric load in reference to time. However, identifying accurate time series models is a challenging task, as it requires training several models to select the best among them. In [8], a linear regression model was used

to perform short-term load forecasting. It classified the load on an hourly basis to model the load for the entire following day. A simple hour-by-hour relation was established using MLR such that the model was developed with a simple least square for general order correlation with a lag load. The lag load tends to have relatively high importance in comparison to other factors affecting the demand. It was not just the regression but the ensemble models that emphasized time lags as the most important feature in comparison with the environmental variables [17]. In the case of France, the model developed showed that the previous 10 days of load data held the highest importance, followed by the temperature and other deterministic variables. When studying the day-to-day load variation, the weekdays and weekends had a visible impact on electricity demand. Studies showed that the demand for weekdays was stable in comparison with the weekends [27]. To better track the effect of special days, the authors of [9] classified the data into three categories: a Sunday and holidays group, a Monday to Friday group, and a Saturday group. With such classification of the load data to different time sets, the forecast obtained an appealing MAPE of 1.17% for the Sunday and holidays group, 0.62% for the working days, and 0.83% for the weekends. The short-term load forecasting methods have a limitation on their use when it comes to special days, weekends, or holidays. One study [10] used a hybrid approach of ANN-based methods and fuzzy logic to forecast the hourly load of special days, as a load on a special day tends to spike up or stay high, load addressing for these days becomes important. Such separation, as well as interactive coupling with other environmental factors, can reduce the prediction error to 2% [10,29].

## 3. Data Analysis

Figure 1 represents one year of hourly load demand data for a Baneshwor substation in Kathmandu, Nepal. The data were collected from the start of the Bikram Sambad year (2075 B.S.), which falls within 2018/2019 A.D. A total of 8760 data were presented for each hour of the year, which ranged from the 14 April 2018 to 13 April 2019 A.D. The demand for the selected feeder line ranged from 0.1 to 2.9 MW. The energy demand of the area followed a cyclical pattern, which could be related to the residential and commercial activities of that area. Based on the demand graph, demand was almost constant from April to September with little variation. During mid-October, a sudden drop occurred, as can be seen in Figure 1b, as did spikes during January and February, as can be seen in Figure 1c.

The box plot of the load consumption in Figure 2 reveals the variation of load for each month throughout the year. The demand tended to spike in the ninth (Paush) and tenth (Magh) months of the year. Here, the ninth and tenth months for B.S. fell in between mid-December and mid-February. The winter season in Nepal starts from the eighth month of the year and lasts until the eleventh month of the year. This indicates that the use of heating load in the winter season increases the demand throughout the season. Likewise, electricity demand can be highly related to the festive season; the major festivals celebrated vary each year in Nepal. For the year 2075 B.S., the demand in the seventh (Kartik) month was noticeably lower than in the other months, as this month of the year hosted the major festivals celebrated in Nepal (Dashain and Tihar), and during that period, all of the commercial businesses, offices, and industries remained closed. Thus, demand decreased significantly during that time of the year, which explains the sudden drop in load in mid-October in Figure 1.

Weekends and weekdays also affect the level of energy consumption. The patterns on weekdays and weekends were quite different. Based on the literature, the demand tends to be higher on weekends than on weekdays in France [17]. For the development of a model, this can be used as an important feature to improve accuracy. However, in the case of Nepal, Saturday is the weekend, while Sundays are the first day of each week. The weekly demand box plot in Figure 3a shows a constant average demand throughout weekdays, as Sunday to Friday are the working days in Nepal. The weekend, the only holiday of the week, had slightly lower demand compared with other days. In Figure 3b, '0' indicates a weekend or holiday, and '1' indicates the working days, showing that the area

being studied had a higher number of commercial and industrial loads, as demand was active on working days. The boxplot shown in Figure 4 presents the daily schedule of the active working hours for the selected area. In observing the hourly variations of electricity demand, the load from 12 a.m. to 5 a.m. had low variation, the demand increased from 6 a.m., and the average demand stayed constant throughout the day until 5 p.m. The demand further increased from 6 p.m. to 9 p.m. during nighttime hours. The electrical demand during nighttime hours was high in these areas, which indicates residential load, utilizing electricity for cooking and lighting purposes.

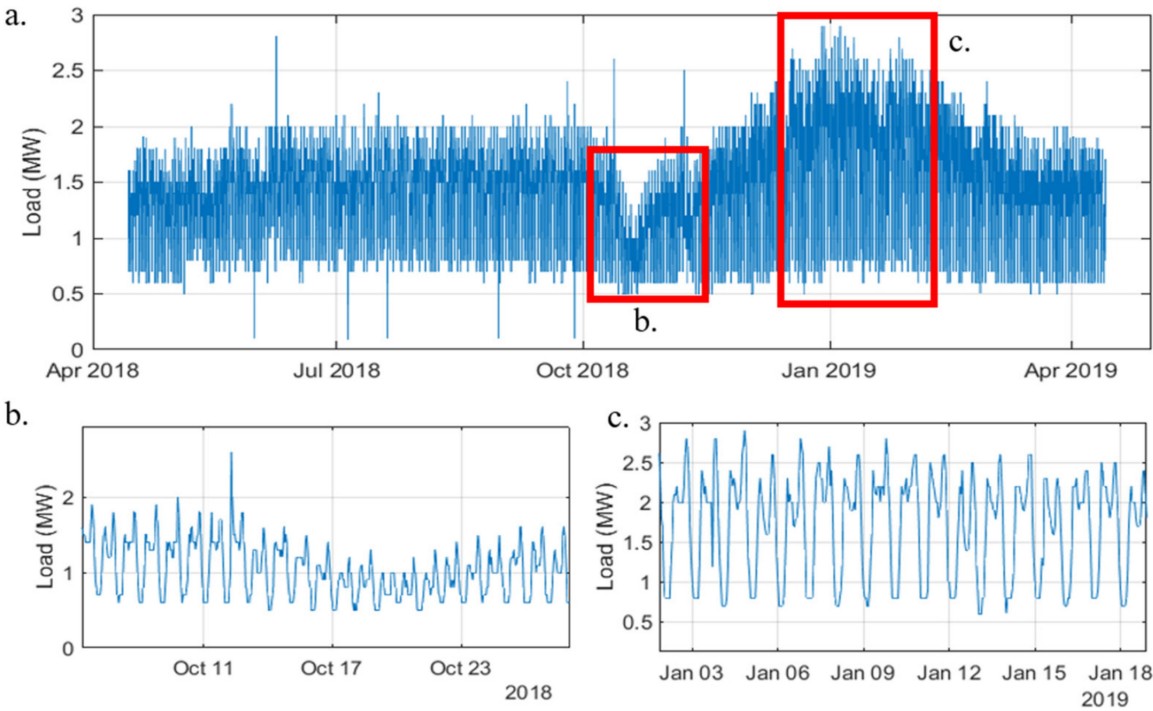

**Figure 1.** (**a**) One year of hourly load demand at a Baneshwor substation (from April 2018 to April 2019). (**b**) Zoomed in section of electricity consumption in October (lower consumption). (**c**) Zoomed in section of electricity consumption in January (high consumption).

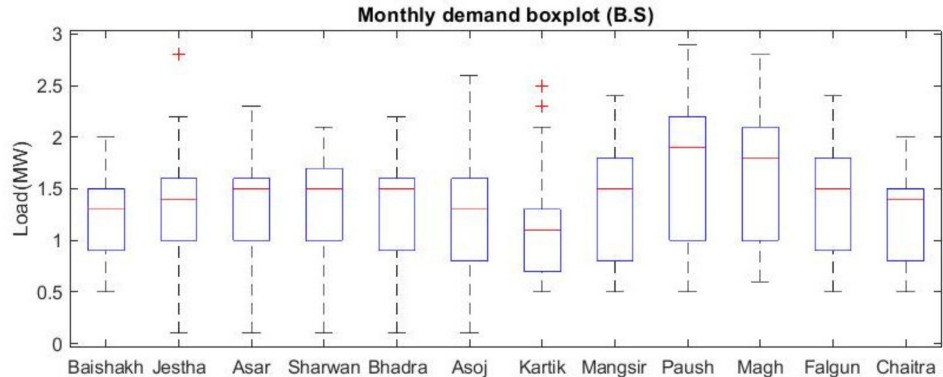

**Figure 2.** Boxplot of electricity demand for each month (B.S.).

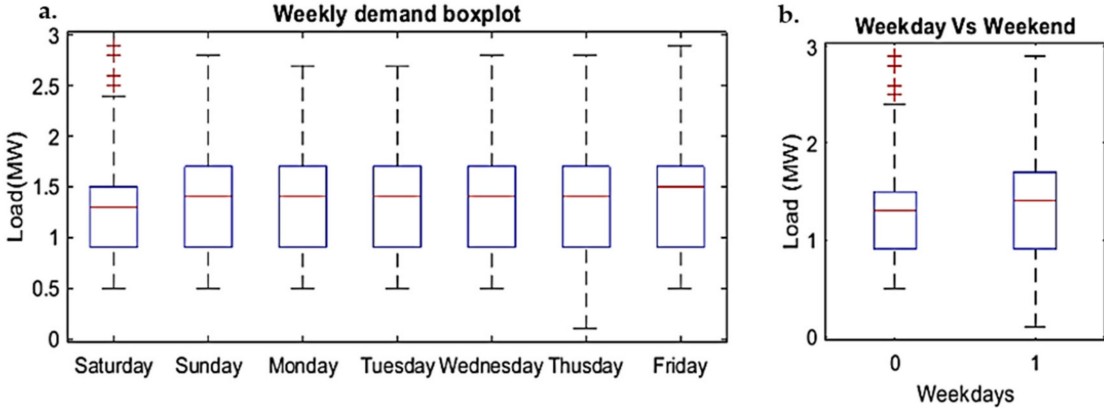

**Figure 3.** (**a**) Boxplot of electricity demand for each day. (**b**) Weekdays vs. weekends.

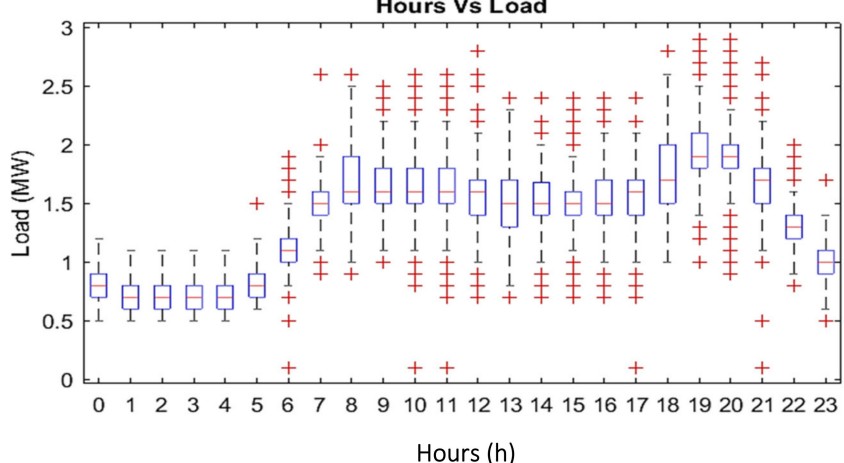

**Figure 4.** Box plot of electricity demand for each hour.

### 3.1. Temperature's Effect on Load

Multiple studies show that electricity demand is highly affected by variations due to various environmental factors [22,24,30]. Among multiple indicators, this paper focuses on load variation due to temperature. Figure 5a shows the relation between the electricity demand and temperature for the case being studied. From Figure 5b, a seasonal variation of the temperature and its effect on the load can be observed. Based on the plot, the positive as well as the negative effects on the load can be observed. From the first month of the year to the fourth month, the temperature had a positive impact on the load. As summer in Nepal starts in late March and lasts until early October, this indicates an increase in the use of cooling appliances as the mean temperature reached 25 °C, with maximum temperatures close to 35 °C. This showed the first spike in electricity demand up to 1.4 MW on average in the area. Likewise, when the year entered the winter season with a mean temperature of 2.5 °C, a second spike in demand occurred of up to 1.7 MW on average. Thus, the minimum temperature could go below 0 °C, and higher use of heating appliances could be observed throughout that period. For observing a mean based correlation plot of rgw monthly mean temperature with the mean load, Figure 5c shows a nonlinear parabolic relation between them. Based on the relation plot of rgw temperature and load, Baneshwor has a base temperature region between 10 and 15 °C. Such an increase or decrease in temperature from this region triggers a rise in electricity demand. The base temperature allows for the calculation of the hot degree days and cold degree days [31], and a study conducted for the load in South Korea showed the difference in sensitivity of the load with hot degree days and cold degree days. The results showed that the load response was larger during hot days than during cold days. The variation in electricity demand around the

base temperature shows the dependency of the load on the air temperature. A nonlinear regression plot of the electric load of Baneshwor showed a parabolic nonlinear relation with the temperature, as can be seen in Figure 5a. In observing the change in demand below and above the base temperature region, the cold temperatures had a greater effect on the load than the hot temperatures. The rate of demand variation per degree of temperature was 0.07 MW/°C during the cold degree days and 0.02 MW/°C during the hot degree days. Thus, the sensitivity of the electricity demand with respect to the temperature was greater in the winter than in the summer. This is due to the power demand gap between heating and cooling appliances, since in developing countries, heaters are the common mode of heating, and fans are used for cooling purposes; very few households rely on air conditioning systems to maintain their indoor temperatures. Similarly, every household owns their own personal boiler and electric geyser for hot water, which further increases the load.

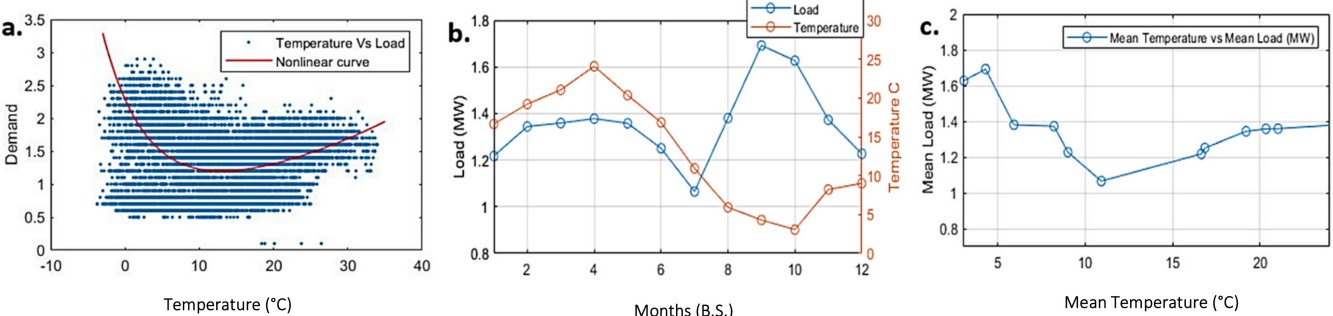

**Figure 5.** (**a**) Joint plot of the electric load vs. temperature. (**b**) Monthly mean plot of the electric load and temperature. (**c**) Mean plot of the temperature vs. electric load.

### 3.2. Time Lag Effect

The lag load or loads from the previous day of the same hour can be highly related to future loads. Studies on lag load relation correlation [8,32] showed that first-day lag impact has a high contribution to the demand. Likewise, one study [7] discussed the repletion of weekdays such that the 7-day lag had a high contribution because of the same day and same hour relating similar effects each day. The correlation plot in Figure 6 shows a high correlation between the electricity demand and the previous time lags, which suggests that the previous day's load of the dependent variable (i.e., electricity demand) could be a useful variable for the model's development.

From Figure 6, we can observe correlations of 0.928, 0.917, 0.912, 0.907, 0.901, and 0.904 with the lag loads from days 1, 2, 3, 4, 5, and 6. The correlations tended to decrease when moving further back in time, which indicates that the load had a similar daily load pattern throughout the week, as short load lag relation could also give satisfying forecasting results. Similarly, Figure 7 shows the correlation plot of the load with the previous 30 lags, in which the correlation value decreased when moving back in time. However, a small spike in the lag on days 7, 14, 21, and 28 can be seen, as these days addressed the similar daily effect such that each weekday repeated after 7 days, with the seventh day's lag being the same for the repeated day of the week. The load showed a 0.9095 correlation value with the 7-day lag load, which was higher than the correlation with the lags for days 4, 5, and 6.

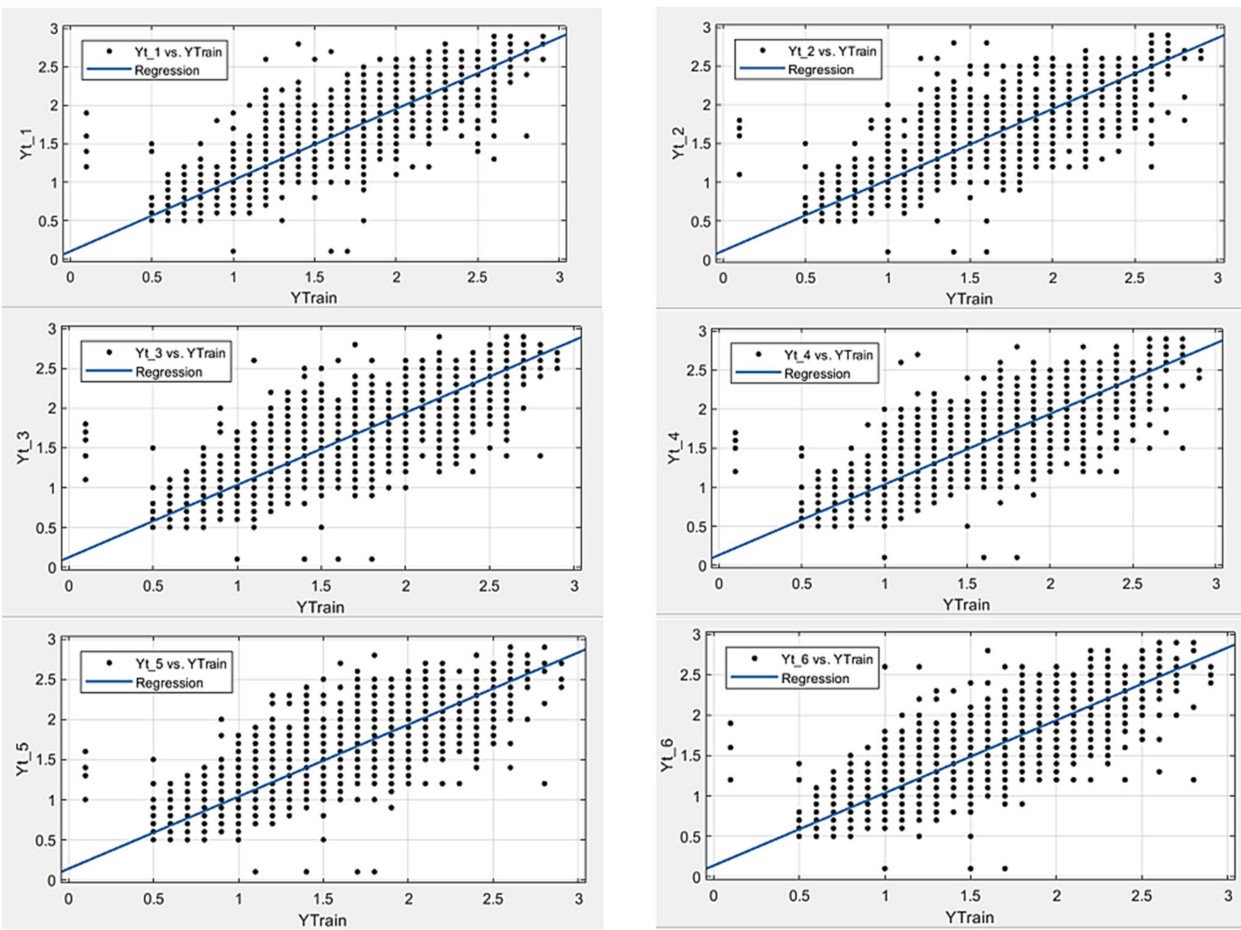

**Figure 6.** Correlation with the previous six load lags.

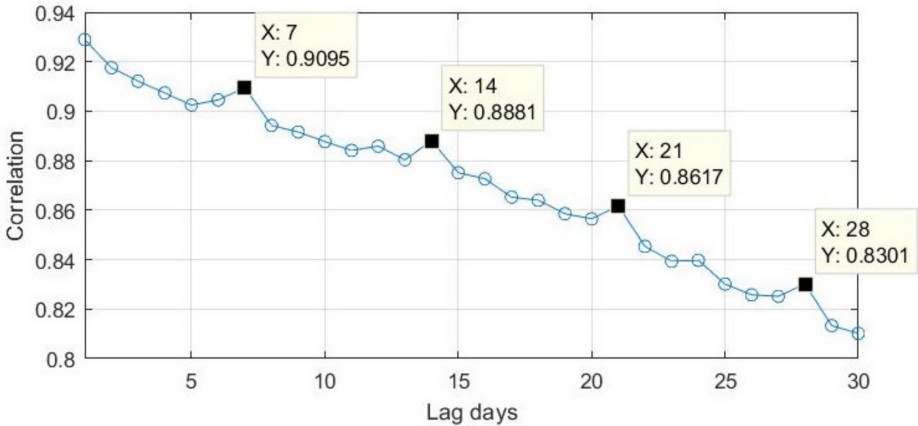

**Figure 7.** Correlation plot for 30 lag days.

## 4. Methodology

In this section, the adopted methodology for short-term load forecasting using the different models is discussed. Figure 8 shows the basic flow of the methodology used for forecasting. The method includes four major steps: (1) the collection of data, (2) the analysis and selection of the data and factors affecting load, (3) model development using time series data for a time series model, and (4) FF-ANN result evaluation and model adjustment. The historic data were thoroughly analyzed to identify the factors affecting the change in demand as discussed in Section 2. The selected data were used to develop the model with multiple iterations and adjustments in the parameters of the model. In our case,

the data were studied and selected for the time series model and compared with a final FF-ANN model. Each model was tested for accuracy and made acceptable by adjusting the model parameters under unacceptable scenarios.

The electricity demand data were observed every hour, and hence there were 24 observations in a day. With the collection of one year of data (i.e., 8760 total observations) as discussed in Section 2, the data were subdivided into 365 sets for each day. Based on analysis performed on the data, they showed a high correlation with the lag load, as benchmark forecasting was performed using various conventional time series methods and a comparative analysis conducted with the proposed ANN model. Initially, the data were fed to conventional time series models and forecasting was performed for two different days: weekends and weekdays. The load showed good correlation with the lag data from Figure 6. Hence, models such as the moving average, weighted moving average, exponential smoothing, and Holt's model, which used lag relation, were used to set the benchmark for our model.

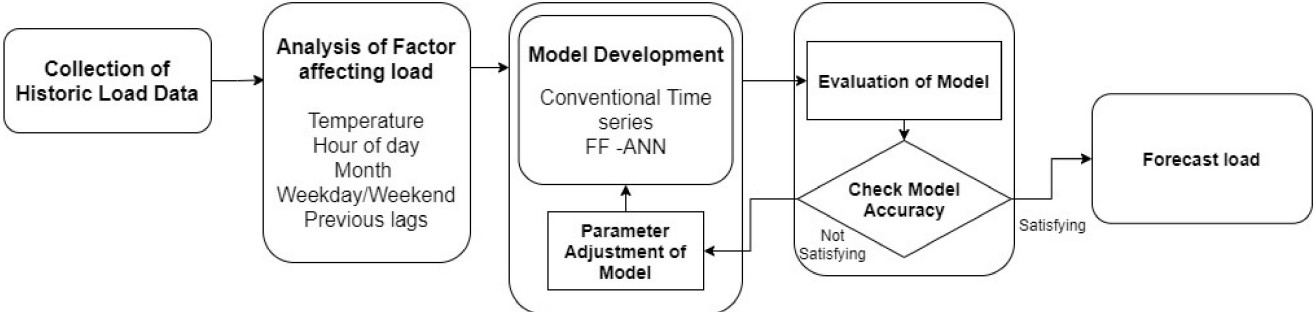

**Figure 8.** Forecasting methodology.

*4.1. Moving Average*

The moving average is one of the simplest ways to perform the prediction of time series data. With the data having a high effect on the previous data, this method became conventional as well as the easiest way to perform forecasting. As the name implies, the forecasted data were the average of the previous lags of data considered (i.e., the 5-day moving averages were the averages of the previous five lags' data). This could be mathematically represented as given in Equation (1):

$$MA = \frac{1}{n} \times \sum_{t=1}^{n} Y_t \tag{1}$$

where $Y_t$ is the data (load data in our case) at time lag $t$ and $n$ is the number of lags considered for $n = 5$ previous data of the same period used to measure the moving average.

*4.2. Weighted Moving Average*

Unlike the simple moving average method, the effect of previous lags on the future load may fade with time. The recent values could have a greater effect on the future values than the past ones. To eradicate the effect of past data on forecasting, varying weights could be assigned to past data to improve the forecasting of the datasets. The weights are assigned the following relations:

$$WMA = \sum_{t=1}^{n} W_t \times Y_t \tag{2}$$

$$\sum_{t=1}^{n} W_t = 1 \tag{3}$$

where $W_t$ is the weight assigned to the lag data $t$. The sum of the weight assigned must be equal to one, as is expressed in Equation (3). Variations in the weight assigned to the formula affects the output of the model. A small shift in the parameter value could result in a huge change in output. Hence, implementing a search algorithm could help save time such that running the model multiple times for variations in the weight values could be achieved.

### 4.3. Exponential Smoothing

The time series forecasting method for univariate data can be extended to support the systematic trend or even the seasonal effect on the system. Exponential smoothing is similar to the weighted moving average in that it assigns an exponentially decreasing weight to past values as the observation gets older. Based on the selected smoothing factors, the effect of an older observation stays in the forecasting. Recent observations are given relatively more weight in forecasting than older observations. The forecasting is performed using Equation (4) [16]:

$$Y\prime_{t+1} = \alpha \times Y_t + (1 - \alpha)Y_t'$$ (4)

where $Y_{t+1}'$ is the forecasted data using actual previous data $Y_t$ and $Y_t'$, which are the forecasted data of previous hours, and $\alpha$ is the smoothing factor with a range of $0 \leq \alpha \leq 1$. As the value of $\alpha$ increases, the effect of the previous value decreases, and vice versa.

### 4.4. Holt's Method

Holt's method is viewed as an extension of exponential smoothing which adds a trend component in the forecasting [33]. Unlike the exponential smoothing method, two components are updated in each interval of forecasting: the level and the trend. The level is a smoothed estimate of the value of the data at the end of each period, while the trend is a smoothed estimate of the average growth at the end of each period. The specific formula for Holt's method is given in Equation (5):

$$Y_{t+1}' = L_t + T_t$$ (5)

where $Y_{t+1}'$ is the forecasted data, $L_t$ is a measure of the change in the level of the forecast, $T_t$ indicates the trend in the data at time $t$, and a sum of the level and trend is used to forecast the date's load on the next period. Equations (6) and (7) present the level calculation and trend calculation of the time series data:

$$L_t = \alpha * Y_t + (1 - \alpha) \times L_{t-1} + T_{t-1}$$ (6)

$$T_t = \beta(L_t - L_{t-1}) + (1 - \beta) \times T_{t-1}$$ (7)

where $Y_t$ is the actual data of the previous lag and $\alpha$, and $\beta$ are the smoothing parameters for the time series data for the limits $0 \leq \alpha \leq 1$ and $0 \leq \beta \leq 1$. In a mode, $\alpha$ controls the weighted average to estimate the level effect and $\beta$ controls the smoothing effect of the trend on the forecast. The simplest way to find the optimal value for $\alpha$ and $\beta$ was using the sum of the squared error of the previous step's prediction [34].

Implementing seasonality can improve the forecasting of the load, as in the case of electricity demand, the seasonal characteristics only repeat after one year. In the case study, the forecasting was conducted using one year of lag data for the seasons. The repetition of data was not available due to a lack of historical data. Hence, Holt's winter method (i.e., triple exponential) for forecasting was avoided.

### 4.5. Feedforward Artificial Neural Network

An FF-ANN is largely composed of three layers, as shown in Figure 9: input, hidden, and output layers. The input layer contains the input variables, which are preselected based on their relation to the output. Selection of the input variable is an important process, as an incorrect selection could lead to overfeeding or underfitting the output. Likewise, the

output layer represents the output obtained from the network or the results of the FF-ANN. The hidden layers consist of neurons which can be designed with multiple hidden layers and have the number of neurons in each layer changed. Some heuristic methods such as particle swarm optimization (PSO), a genetic algorithm (GA), and a combination of both are used to decide the number of layers by finding the global minima based on an error in forecasting [15]. However, in most cases, the selection is performed through a gradual increment in the number of layers and neurons and compared with the error [7]. In Figure 9, the *n* represents the number of hidden layers. Each layer consists of neurons connected by a weighted channel, which represents the weight of the data passed, and $w_n$ presents the weight assigned to every connection.

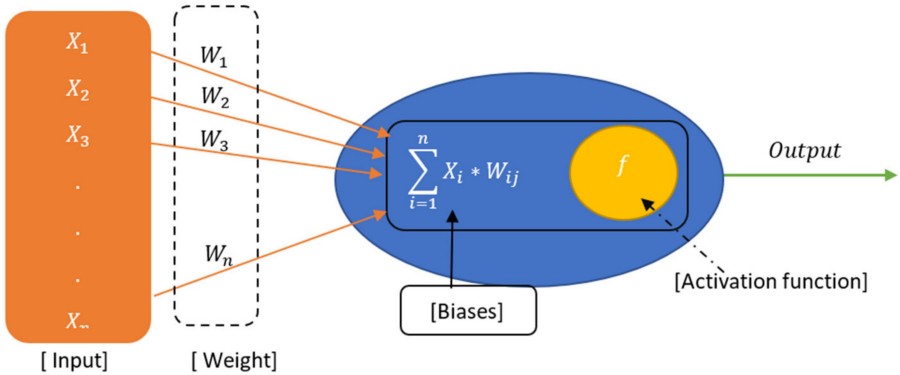

**Figure 9.** Feedforward neural network structure.

At each neuron, data are passed through an activation function (e.g., log-sigmoid, sigmoid, or tan-sigmoid) which determines if each particular neuron is activated or not. Based on the activated neuron, the data are further transmitted to the next layer of the neuron's overweighted channel. The transmission of data is in a forward direction and hence is called the forward propagation of the network. The network is a model in two phases: training and testing. During the training phase, each output is compared with the actual output to realize the error, which works as feedback to the network in the second iteration. The process of training the network is known as backpropagation. Backpropagation helps to adjust the weight based on the given error obtained between the target and the output [15]. It is a form of supervised learning in which the input is repeatedly presented to the network with an error from the previous iteration. The input signal going to each node can be presented as in Equation (8):

$$V_j = \sum_{i=1}^{n} X_i \times W_{ij} + B_j \tag{8}$$

where *i* represents the number of inputs, *j* represents the number of hidden layers, $X_i$ represents the *i*th input, and $W_{ij}$ represents the weight channel from the *i*th input node to the *j*th hidden layer. The signal passed through the activation function which decides the activation of the node is presented in Equation (9). The hyperbolic tangent sigmoid transfer function is used as an activation function to train the network:

$$Y_j = \varnothing\left(V_j\right) = \varnothing\left(\sum_{i=1}^{n} X_i \times W_{ij} + B_j\right) \tag{9}$$

where $\varnothing$ is the activation function and $Y_j$ presents the output from each node. Similarly, each input signal is passed throughout the defined number of nodes, and the final output is obtained as result. In the case of supervised learning, the method in every weight iteration is updated. Among multiple backpropagation learning methods, the Levenberg–Marquardt algorithm is designed to minimize the sum of the sure error function to update the weight

in the ANN. This backpropagation method is considered the fastest learning method in MATLAB for simulations [35]. The updated weight is presented in Equation (10):

$$W(j+1) = W_j - \left[ J^T \times J + \mu \times I \right]^{-1} \times J^T \times e \tag{10}$$

where *J* is the Jacobian matrix that contains the derivative of the error calculated using the initial iteration of weights and biases and *e* is the error calculated for each iteration. The equation presents a combination of Newton's methods and gradient descent. When μ is near zero, the weight is updated using Newton's method, and the larger value of μ in the equation acts as a gradient descent algorithm with smaller steps. The Levenberg–Marquardt algorithm aims to shift from gradient descent to Newton's method after each successful step.

### 4.6. Performance Evaluation and Validation

One of the more common methods used to evaluate the performance of the forecasting model is MAPE. It measures the deviation of the predicted data with the real values in terms of a percentage:

$$MAPE = \frac{1}{N} \sum_{n=1}^{N} \left| \frac{Actual_n - Predicted_n}{Actual_n} \right| \times 100\% \tag{11}$$

It measures the deviation of the overall data set, where *N* represents the length of the data set predicted in the future. In our case, forecasting was performed for the next 24 h.

## 5. Results and Discussion

### 5.1. Conventional Time Series Model Testing and Analysis

To obtain the best result, the parameters were selected based on the minimum MAPE for each forecasted value. For the initial input for the time series model, the time lags were the complete set of lag values. Table 1 shows the parameters for the selected model.

**Table 1.** Parameter selection of the time series model.

| No. | Model | Parameters (Weekdays) |
|-----|-------|------------------------|
| 1 | Moving Average | Lag load = 4 (4-day lag) |
| 2 | Weighted Moving Average | Lag Load = 4 $(W_1 = 0.4,\ W_2 = 0.3,\ W_3 = 0.2,\ W_4 = 0.1)$ |
| 3 | Exponential Smoothing | $\alpha = 0.15$ |
| 4 | Holt's (Double Exponential) | $\alpha = 0.14,\ \beta = 0.08$ |

Based on the lag load, considering that the error varied for the MA and WMA models, a maximum of 100 days of lag was considered for analysis, and the minimum MAPE was obtained when the model considered the previous values of 4 days. Averaging a higher lag value did not affect the MAPE. In the case of the WMA model, considering the MA model as a benchmark, the model considered 4 days for the maximum lag and weight. The WMA model was run through a simple exhaustive search algorithm of $10 \times 4$ iteration points to find the 4 optimum weight values. The variation was only performed for one decimal point so that the sum of the weight equaled one, as dictated by Equation (3). Likewise, for exponential smoothing and Holt's method, the modeling again followed an exhaustive search algorithm to find the optimal value. The search ranged from 0 to 1 with 2 decimal points (i.e., a total of 100 iterations was performed). From the parameters selected from the MA, WMA, and Holt's method models, it was indicated that previous values had a lower effect on the forecasting. Similarly, the value of beta was lower, indicating no effect of the trend between the lag values.

Analysis of the time series forecasting method was conducted using forecasted data for two days, as the study on the data indicated variation in the load demand for weekdays

and weekends. The first forecasting was performed for a normal weekday (12 April 2019), specifically Friday, and the second was performed for a weekend (13 April 2019) on a Saturday. From Figure 10a,b, it can be seen that the models could maintain the pattern of the load. From 12:00 a.m. to 7:00 a.m., the load had a similar pattern for both weekdays and weekends. However, the electricity demand during the daytime from 8:00 a.m. to 9:00 p.m. was comparatively lower on weekends than on weekdays such that the model performed better for working days.

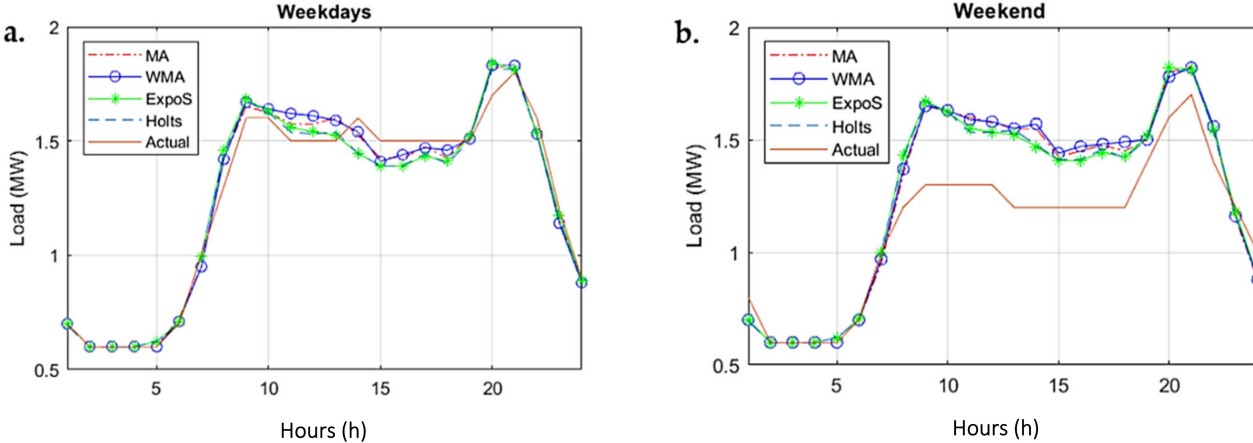

**Figure 10.** Actual vs. predicted forecasts achieved by conventional time series models. (**a**) Weekday forecasting. (**b**) Weekend forecasting.

The accuracy in the forecasting of the weekday load and weekend load showed a huge difference for all the time series models. Table 2 shows the MAPE (%) of each time series model. Here, each model archived satisfying results for the weekdays. The comparison shows that the MAPE achieved for the MA model was comparatively lower than the other models for working days. The results indicate that the load on each working day was similar to the previous day's load. However, when observing the MAPE on the weekends, the best result obtained was 12.57% for Holt's method. Even though the system was able to grasp the daily load pattern as observed in Figure 10b, the average load demand on the weekend was comparatively lower, setting the model forecast value off the course during the weekend. Likewise, with lower values for alpha in both the exponential model and Holt's method, the effect of the previous load lags was lower. This shows that higher order lag had a lower influence on the current load demand.

**Table 2.** MAPE comparison of time series models.

| No. | Model | MAPE % (Weekdays) | MAPE % (Weekends) |
|---|---|---|---|
| 1 | Moving Average | 3.33 | 13.42 |
| 2 | Weighted Moving Average | 3.4 | 13.66 |
| 3 | Exponential Smoothing | 3.56 | 12.59 |
| 4 | Holt's (Double Exponential) | 3.44 | 12.57 |

*5.2. Feedforward Artificial Neural Network Parameter Selection and Result Analysis*

A feedforward neural network was developed to predict the load for the considered days. The input variables were designed from the MATLAB environment using the MATLAB 2018 Neural Networks Toolbox. Model training, testing, and validation are three major parts of ANN modeling, and each dataset is divided into three layers: the first 70% of the data is used to train the data, 15% for the test, and 15% for validation.

In the initial time series models, only time series data were considered, ignoring the effect of a nonlinear temperature and similar day effects, while the FF-ANN model

additionally considered the temperature data to catch the non-linear relation between the temperature and the relation to similar day effects such that each day was separately marked, including variables like the month, day of the month, day of the week, hours of the day, same day of the previous week, and flag indicators for weekdays and weekends. The variables selected can be seen in Table 3.

**Table 3.** ANN input variables.

| No. | Variable | Indicator |
|---|---|---|
| 1 | Month | 1, 2, 3 . . . , 12 |
| 2 | Day of the Month | 1, 2, 3, . . . , 30, 31 |
| 3 | Week | Sunday-1, Monday-2 . . . Saturday-7 |
| 4 | Hour of Day | 0, 1, 2, 3 . . . , 23 |
| 5 | Weekday, Weekend | 1, 0, |
| 6 | Previous Day's Load | L1(t-24), L2(t-25) . . . L1(t-47) |
| 7 | Same Day of the Previous Week | L1(t-24 $\times$ 7) . . . L24(t-47 $\times$ 7) |
| 8 | Temperature | T(t) |

A network was trained with the historical data and other factors considered. The variables were assigned with numerical values as indicated in Table 3. These allowed the ANN to trace similar day effects. In the case of the months, the calendar considered the B.S. calendar; hence, 1 indicated the first month of the B.S. calendar, which is 'Baishakh'. The test was performed for various structures, consisting of up to three hidden layers with different nodes as discussed in Table 4. The same test data were considered for both weekdays and weekends. Considering the difference in nature of the demand on weekends and weekdays, two different networks were modeled. Multiple nodes and layer structures were tested, and from Table 3, it can be observed that a single-layer structure with eight nodes showed the best MAPE of 2.99% for weekdays, and a two-layer structure with 16 $\times$ 16 nodes presented the best MAPE of 4.69% for the weekend load forecasting.

**Table 4.** Selection of the hidden layer.

| No. of Hidden Layers | No. of Neurons | MAPE % (Weekends) | MAPE % (Weekdays) |
|---|---|---|---|
|  | 32 | 4.89 | 3.22 |
|  | 16 | 4.699 | 3.14 |
| 1 | 8 | 5.08 | 2.99 |
|  | 4 | 6.07 | 3.04 |
|  | 2 | 7.16 | 4.77 |
|  | 1 | 13.38 | 12.08 |
|  | 2 $\times$ 2 | 7.06 | 3.96 |
| 2 | 4 $\times$ 4 | 4.76 | 3.26 |
|  | 8 $\times$ 8 | 4.76 | 3.63 |
|  | 16 $\times$ 16 | 4.53 | 3.04 |
| 3 | 8 $\times$ 8 $\times$ 8 | 4.90 | 3.39 |
|  | 16 $\times$ 16 $\times$ 16 | 4.54 | 3.35 |

From Figure 11a,b, it can be seen that while training the model using the Levenberg–Marquardt backpropagation algorithm, the model converged with less than 20 epochs for weekdays for the model and the single hidden layer having 8 neurons, and it also converged before 16 epochs for the weekend, which was modeled with the double hidden layer having 16 $\times$ 16 neurons. The implemented model configuration showed no overshoot before or after converging and maintained stability (no increase or decrease after converging), with its best validation performance at epoch 12 and epoch 10 for the test dataset.

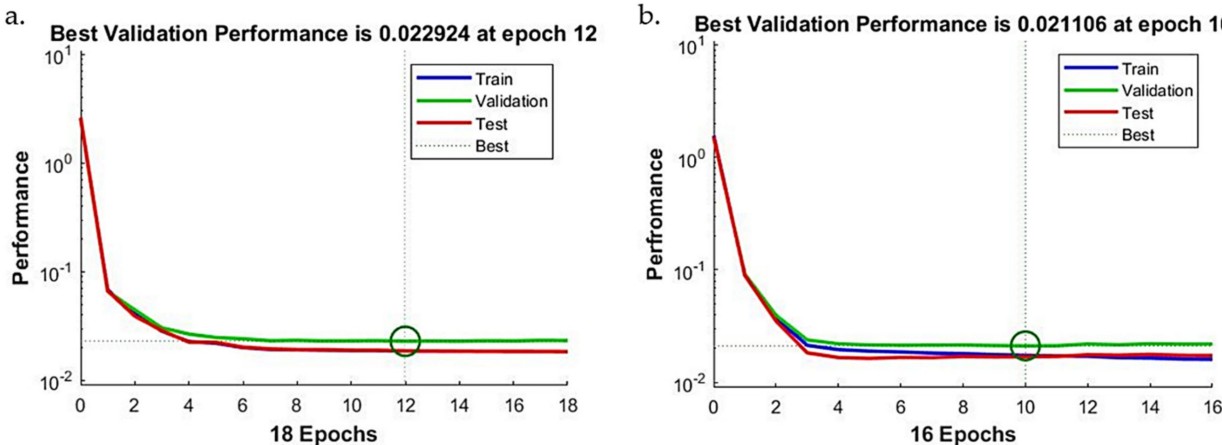

**Figure 11.** Training performance graphs for (**a**) weekdays (**b**) weekends.

Unlike the time series models, the ANN considered all the parameters so that a higher number of parameters did not affect the forecasting; it tended to ignore the unnecessary terms as it progressed. Comparing the ANN model with the time series forecasting models, the MAPE on the weekdays was improved by 0.34%, and for the weekends, it improved by 8.04%. The time series models were unable to track the weekend demand pattern, and the magnitude of the load demand was way off course during the daytime from 8:00 a.m. onward. On the other hand, the FF-ANN model performed much better in terms of tracking the load for both types of days. As can be seen in Figure 11, in terms of the pattern as well as the magnitude of the electricity demand, the FF-ANN was able to follow both load profiles. When comparing the predicted data with the load forecasting using time series, especially on the weekend, we can observe a high variation in load amplitudes in Figure 10b. With the ANN implemented, a huge improvement can be observed for the weekend from Figure 12b.

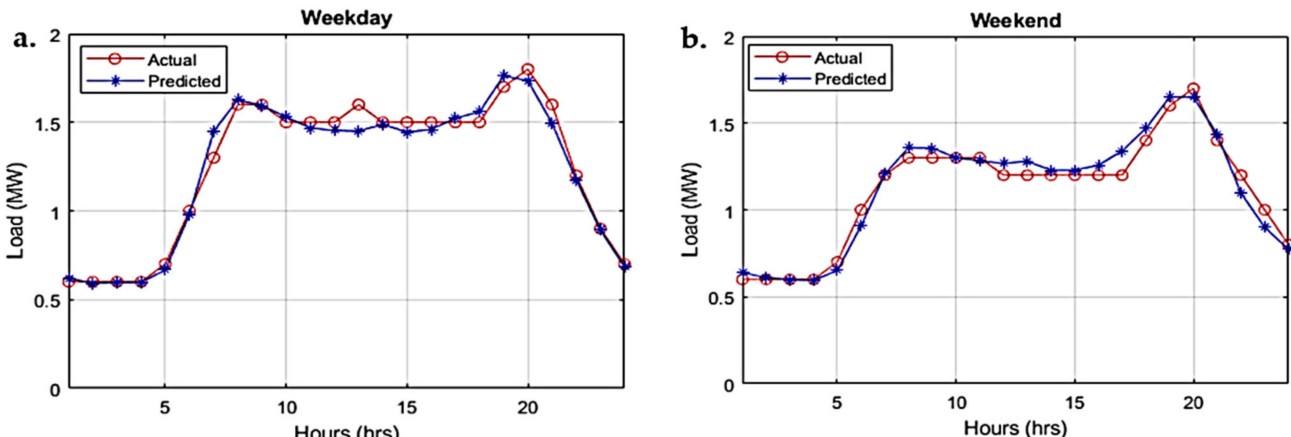

**Figure 12.** Actual vs. predicted forecasts achieved by the ANN models with (**a**) a double-layer 16 × 16 node structure and (**b**) a single-layer 8 node structure.

## 6. Conclusions

With the growing electricity demand, the power system in Nepal is moving from being supply-based to demand-based, where the stability of the system depends on the system's capacity to handle variations in load demand. Being a developing country, the power system has not tackled issues related to demand variation, and high power fluctuation due to variations in demand could lead to power system issues, thus leading to system instability due to unsolved unit commitment problems and resulting in voltage fluctuation, frequency fluctuation brownout, or even blackouts. As a sudden spike in electric load

requires a quick response and high storage capacity in the power system, developed countries rely on forecasting electric load consumption to track the variation in load and identify the unit commitment problem that the system might face in the future. Utilities develop forecasting models that are a day ahead, a week ahead, or a month ahead to allow them to plan generation and, in many cases, implement demand management strategies such as the timed of use tariffs, incentive-based demand response, or even direct control of load to maintain stability in the system. With technological advances on the demand side, policies and strategies based on forecasted load such as critical pricing can help reduce peak load consumption. As barely any research has been performed on load or on electric load consumption in Nepal, this presented study allows the utility to understand the characteristics of the load and its variation due to variables that affect the electric demand of the urban cities of Nepal.

This paper presented load forecasting models based on the relation of load with lag loads and temperature for the urban cities of Nepal. Based on the performed correlation study, high correlation with previous lag loads showed that the conventional time series forecasting method could achieve satisfying results for normal working days, whereas the model developed lacked accuracy on weekends such that correlation tended to decrease when moving further back in time, showing limited lag, which could be considered to maintain its accuracy by using the time series model. An improvement in accuracy for load forecasting was achieved by using an FF-ANN through the thorough analysis and selection of input variables. By analyzing the variables affecting the load, it was observed that the variation of electricity demand with the temperature showed both positive as well as negative effects on the demand. For the first six months of the year, the temperature had a positive impact on the demand, which indicates that the use of cooling appliances rose during that period since the mean temperature reached 25 °C, increasing demand to 1.4 MW in the summer. Similarly, in the next six months, the average temperature decreased to 2.5 °C, which spiked the demand to 1.7 MW in the winter season. The sensitivity of electricity demand to the temperature was larger in the winter than in the summer, as the rate of demand variation per degree of temperature was observed to be 0.07 MW/°C during the cold degree days and 0.02 MW/°C during the hot degree days. The daily load demand and weekly variation showed the daily load activity pattern and type of user. As demand increased from 6:00 a.m., the average demand stayed constant throughout the day until 5:00 p.m. Then, the demand stayed high from 6:00 p.m. to 9:00 p.m. This study indicated the variation of load on weekdays and weekends such that a single model was not able to analyze both cases; a different model was required to forecast the demand on the weekends. By including the temperature and similar day effect as additional variables of the load fed to an FF-ANN model, the developed model successfully achieved a minimum MAPE of 2.99% for the weekdays and 4.53% for the weekends. The comparison showed that the MAPE obtained on the weekdays was improved by 0.34% and 8.04% for the weekends with an FF-ANN with respect to the conventional time series models.

From this study, it was concluded that an FF-ANN is an acceptable forecasting model for both weekdays and weekends when compared with conventional time series load forecasting models with proper input variables available. However, in the case of a lack of data on the influencing variables, the time series model itself can be used to forecast the electricity load for weekdays, whereas a different approach for forecasting should be adopted for weekends. Conventional time series methods lack flexibility in addressing the effect of exogenous variables, which is a major drawback to the time series model, especially when addressing weekends, holidays, days of the month, and temperature, which all hold important information on the electric load pattern of the day. The weekdays and weekends resulted in different human activities, similar to special days, public holidays, and festive seasons also resulting in changes in human activities, resulting in changes to the electric load. Even with the ANN model's capability to track nonlinear relations to multi-variable inputs, the model is solely dependent on the input variable selected; thus, input error can result in major error in the performed forecasting. The proposed model is developing the

use of one year of data as it lacks information on special events within the country, but with correct data on the events, the model has further room for improvement. This study can be further expanded upon to understand the relation of the load with the festivals and special events of Nepal, which are a major cause of electric load fluctuation.

**Author Contributions:** Conceptualization, Y.R., A.S. and A.G.; methodology, Y.R., A.M., B.G. and K.C.; software, Y.R. and B.G.; validation, A.S., A.G., A.T., K.C. and P.K.; formal analysis, Y.R., A.M. and B.G.; investigation, Y.R., A.M. and B.G.; resources, A.S., A.G. and A.T.; writing—original draft preparation, All; writing—review and editing, All; supervision, A.G. and P.K.; project administration, A.S. and A.T. All authors have read and agreed to the published version of the manuscript.

**Funding:** This research was funded by the EnergizeNepal Project Office of Kathmandu University, grant number ENEP-RENP-II-19-03.

**Data Availability Statement:** The data will be available upon request.

**Acknowledgments:** The authors are thankful to the Department of Electrical and Electronics Engineering and the Center for Electric Power Engineering (CEPE) at Kathmandu University for providing the requested information, laboratory supports, and kind cooperation during this research work. A special thanks to Binamra Adhikari and the Nepal Electricity Authority Engineering Company of Nepal for sharing the data of the electrical load demand of the Baneshwore.

**Conflicts of Interest:** There is no conflict of interest.

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
