# Peer review of "Impact Study of Temperature on the Time Series Electricity Demand of Urban Nepal for Short-Term Load Forecasting"

_asi, doi:10.3390/asi4030043_

Round 1

Reviewer 1 Report

  • Interesting and less studied region where method is applied
  • The introduction gives a clear overview. More references are presented in the literature review in section 2
  • The paper is well structured
  • The paper is understandably written and the English in general is fine. However there are several minor spelling errors that need to be corrected:
  • First of all in the title, “Time Series” should be written with capital letters
  • Line 19: “it’s” instead of “it”
  • Line 29: In the keywords sometimes commas and sometimes semicolons are used. Also there is a comma at the end
  • Line 78 – 86: Sometimes the list ends with a dot, sometimes not
  • Line 161: Short Term Load Forecasting instead of Short term
  • Line 177: It would be good to explain the abbreviation LSTM in the text
  • Line 208: “during night hours result show”. It looks like a word (e.g. “where”) is missing
  • Line 355: “is discussed” instead of “has been discussed”
  • Line 370: “the data is fed” instead of feed
  • Figure 8: Ensure uniform upper / lowercase spelling
  • Line 393: “assigned to the t lag data” Should it be Yt instead of t?
  • Line 419: Should it be trend instead of tend?
  • Line 463: “ith” instead of “ith”
  • Line 471: “to update the…”
  • Line 473: “is presented instead” of “can be present”
  • Line 477: “Newton” instead of “newton”
  • Please check within the entire paper that “time series” is written in a uniform way. Sometimes it’s written “time Series”, sometimes “time series” and so on
  • Line 497: “it was indicated” instead of “indicated”
  • Line 500: “was done” instead of “as done”
  • Caption of Figure 10 and 11: “Actual vs. predicted…”
  • Table 3: A space is missing after Table 3:, “Day of month” instead of “Day of Month”
  • Table 4: Space after “Mape %” is missing
  • Line 561: “Comparing the…” it looks like a word is missing
  • Line 563: After “Figure 10” it looks like a dot is missing
  • Line 563: I assume it should mean Figure 11 instead of 12. It also looks like a space is missing after Figure 11: in the caption
  • Line 687: There is 2x “Holt’s” in the articles name
  • Line 683: Is this reference complete?
  • Section 2.1: It would be interesting to read more about the reasons of the temperature effect (e.g. chillers, electric heating devices etc.). For Kathmandu valley this is given in the conclusions
  • For future work it could be interesting to also investigate the impact of global radiation on the load. E.g. the increasing number of PV systems could lower demand and also increase the electricity fed into the grid. It could also affect the user behaviour and the usage of cooling devices

Author Response

The author/s appreciate the detailed review from the reviewer and pointing out key elements that the manuscript was lacking. We hope the revised manuscript will satisfy the reviewer. We have addressed all of the comments pointed out by the reviewer except few for following reasons.

Line 393: “assigned to the t lag data” Should it be Yt instead of t?

            t is correct as we are considering t lag Y refers to data for t lags

Line 419: Should it be trend instead of tend?

            Trend is correct: as time series data follows a trend

Reviewer 2 Report

The paper employs an ANN model to track the effect of temperature and similar day patterns by using the conventional time series model as a benchmark. It is generally well written and organized. It targets an interesting subject.

I have a few recommendations:

  • the introduction should better explain how the current paper contributes the literature, underline its extent of novelty, and “put it in context” in the extant literature. As it is, we have no clear indication of the contributions this paper brings and if/why it is important.
  • - The discussion and Conclusion parts are merely technical and could be improved by also offering some economic effects and policy implications. Conclusions should not repeat the technical results, but go further and explain how they differ or concur with previous studies. Policy implications should be underlined; this is probably the most important aspect of the paper and its “raison d’etre”. Some weaknesses of the present study should be acknowledged and future research directions underlined. 
  • - Also, check the following sentence: ” With the study of electricity demand in Baneshwore substation concerning previous days load of the same hour and temperature as an environmental variable affecting the load.” – row 73
  • - Check also the following paragraph (rows 76-86), there are inconsistencies in the way the text is expressed. 

Author Response

Respected sir/ Madam

The author/s appreciates the detailed review from the reviewer and point out important point to improve the quality of the paper. We hope the revised manuscript has address all your comment and will satisfy the reviewer. 

Reviewer 3 Report

Review:

Impact Study of Temperature on Time series Electricity Demand of Urban Nepal for Short Term Load Forecasting

This impact study highlights a data analysis process that should motivate other load forecasting projects regarding the inclusion of additional influential variables such as temperature. It is important to have similar impact evaluations of those variables because the implementations of multivariate forecasting models are often optimized to perform well under use case scenarios that are affected by locality and the overall structure of the available dataset. A few points need to be addressed in a revised version:

  1. Figure 1 could be replaced with a panel of 3 figures. Figure (a) could include the one-year hourly load demand and figures (b) and (c) could include the specified regions highlighted in the original figure in order to avoid overlapping figures.
  2. Since both Table 3 and lines 539-543 explain the input variables and their values, the information provided could be seen as redundant. Therefore, one of the two approaches should be selected and, in that approach, the detailed list of input and output variables should be included with their respective encoding.
  3. Since MAPE is used as the error metric, a proper definition of the metric should be included with the appropriate formula and a sufficient explanation on why this error metric is the most suitable for this forecasting task.
  4. For completeness, you are encouraged to include any additional information that further explain the training process of each neural network model such as number of epochs, the selected optimizer etc. Additionally, you could provide a graph showing the value of the loss function as it decreases through the epochs for the test set in order to see how smooth the path to convergence can be given the proposed neural network configurations.
  5. In the conclusion, it is mentioned that in the case of lack of data a different forecasting approach should be adopted for weekends. It would be beneficial to expand on the challenges/drawbacks of those forecasting models in that last part of the conclusion with a few more lines of text so that future research projects following a similar set of variables could avoid unusual outcomes such as overfitting.

Author Response

Respected sir/madam

The author/s appreciates the detailed review from the reviewer, and thankful for the valuable feedback to improve the quality of the paper. We hope the revised manuscript has addressed all your comments and will satisfy the reviewer.

Thank you
